# Is There a Glutathione Centered Redox Dysregulation Subtype of Schizophrenia?

**DOI:** 10.3390/antiox10111703

**Published:** 2021-10-27

**Authors:** Lena Palaniyappan, Min Tae M. Park, Peter Jeon, Roberto Limongi, Kun Yang, Akira Sawa, Jean Théberge

**Affiliations:** 1Department of Psychiatry, Schulich School of Medicine and Dentistry, Western University, London, ON N6A 5C1, Canada; matt.park@lhsc.on.ca (M.T.M.P.); jtheberge@lawsonimaging.ca (J.T.); 2Department of Medical Biophysics, Western University, London, ON N6A 5C1, Canada; yjeon4@uwo.ca; 3Robarts Research Institute, Western University, London, ON N6A 5C1, Canada; rlimongi@uwo.ca; 4Lawson Health Research Institute, London, ON N6C 2R5, Canada; 5Department of Psychiatry and Behavioral Sciences, Johns Hopkins University School of Medicine, Baltimore, MD 21205, USA; kunyang@jhmi.edu (K.Y.); asawa1@jhmi.edu (A.S.); 6Department of Biomedical Engineering, Johns Hopkins University, Baltimore, MD 21218, USA; 7Department of Neuroscience, Johns Hopkins University School of Medicine, Baltimore, MD 21205, USA; 8Department of Genetic Medicine, Johns Hopkins University School of Medicine, Baltimore, MD 21205, USA; 9Department of Mental Health, Johns Hopkins Bloomberg School of Public Health, Baltimore, MD 21205, USA

**Keywords:** glutathione, glutamate, psychosis, schizophrenia, redox, antioxidant, oxidative stress, myelin, spectroscopy

## Abstract

Schizophrenia continues to be an illness with poor outcome. Most mechanistic changes occur many years before the first episode of schizophrenia; these are not reversible after the illness onset. A developmental mechanism that is still modifiable in adult life may center on intracortical glutathione (GSH). A large body of pre-clinical data has suggested the possibility of notable GSH-deficit in a subgroup of patients with schizophrenia. Nevertheless, studies of intracortical GSH are not conclusive in this regard. In this review, we highlight the recent ultra-high field magnetic resonance spectroscopic studies linking GSH to critical outcome measures across various stages of schizophrenia. We discuss the methodological steps required to conclusively establish or refute the persistence of *GSH-deficit* subtype and clarify the role of the central antioxidant system in disrupting the brain structure and connectivity in the early stages of schizophrenia. We propose in-vivo GSH quantification for patient selection in forthcoming antioxidant trials in psychosis. This review offers directions for a promising non-dopaminergic early intervention approach in schizophrenia.

## 1. Introduction

Schizophrenia is one of the most devastating of adolescent onset illnesses. Despite the advances in pharmacological, psychological, and social aspects of care in the last 50 years, only a small sub-group achieves combined clinical and functional recovery (~13%) [1,2]. Over several decades, barely any improvement has occurred in life expectancy and the years of potential life lost [3]. All the currently available antipsychotics focus on dopamine, but the symptom relief they provide does not translate to functional recovery in most cases. Remarkably, around 10% of patients do not develop further episodes after the first [4]; but in those who experience persistent illness or recurrences, antipsychotics do not reverse the cognitive deficits and negative symptoms that contribute to most of the functional disability [5,6]. Currently there are no truly ‘disease-modifying’ interventions available [7] though many promising leads have emerged in recent times.

While a complete mechanistic account of the schizophrenia or the ‘group of schizophrenias’, as Bleuler surmised 110 years ago [8], is still lacking, three key neurobiological substrates of the disabling illness trajectory have emerged in recent times. (1) glutamatergic dysfunction in early stages [9,10] related to excitation–inhibition imbalance resulting from prefrontal parvalbumin interneuron deficits in the cortical microcircuit [11,12] (2) dysconnectivity of large-scale brain networks (especially involving the dorsal anterior cingulate cortex [ACC] and insula) in prodromal stages before the first-episode [13,14,15,16], related to persistent symptom burden [17,18,19] and cognitive deficits [20,21,22], and (3) microstructural changes in the grey [23,24] and white matter [25,26,27], predating the illness but becoming more prominent during the first-episode [24], possibly reflecting the loss of dendritic spines [28,29] and myelination deficits [30,31]. Despite these mechanistic insights no accessible therapeutic targets have emerged yet. This is because most changes occur in early developmental periods that precede the illness onset by many years. By the time symptoms first appear, most neural disruptions are already well established and often irreversible. Therefore, there is an urgent need to identify pathways of poor outcome in schizophrenia that remain ‘modifiable’ *after* the symptom onset.

## 2. Glutathione in Schizophrenia

One of the pathways of poor outcome that has its roots in early life but continues to remain modifiable in later stages of illness may be the antioxidant pathway. Destructive free radicals that damage brain tissue are by-products of oxidative metabolism but are effectively scavenged by antioxidants. Glutathione [GSH] is the cardinal antioxidant in brain cells. In preclinical models of schizophrenia, early GSH deficit contributes to dysfunction of prefrontal parvalbumin interneurons [32,33], increased susceptibility of excitotoxic pyramidal cell damage especially in the presence of a hyperdopaminergic state [34,35], reduced dendritic spines [36], reduced stability of axonal projections [37], and facilitates oligodendrocyte cell death disrupting myelin formation [38,39,40,41]. Crucially, agents that improve GSH levels ameliorate the effects of oxidative stress in various preclinical models of schizophrenia (ketamine [42], GluN2A [43] or neonatal ventral hippocampal lesions [44,45], perinatal infection [46], stimulant exposure [46], and maternal immune activation [47]). This large body of evidence has led to the claim that various mechanistic strands underlying schizophrenia converge on the “hub of oxidative stress” indexed by GSH [48,49]. Preclinical models with early GSH deficit (i.e., from postnatal day 0 onwards) display various schizophrenia-like features in adult life, including a sensitivity to dopamine excess [50], and prefrontal hypomyelination [51]. Nevertheless, chronic peri-adolescent treatment with the glutathione precursor N-acetylcysteine (postnatal days 5–90) restores antioxidant-related and myelin-related mRNA expression improving cognitive flexibility in later life [51]. Thus, despite the likely developmental origins of the GSH-deficit, a phenotype ‘rescue’ is possible in preclinical models with interventions in later life. Such ‘rescue’ effects on myelination have also been reported in patients with early psychosis taking GSH precursors as supplements [52], offering GSH-deficit as a potentially modifiable pathway of poor outcomes in schizophrenia. In this context, early identification of individuals with GSH-deficit from clinical samples of patients with schizophrenia assumes paramount importance.

## 3. Intracortical GSH in Schizophrenia: MRS Studies

Peripheral antioxidant markers are reduced in patients with schizophrenia [53,54]. Genetic [55,56] and cell biology studies [57,58,59] indicate that the ability to produce GSH in the face of oxidative stress is likely to be reduced in the patients, at least a subset of them. More direct demonstration of low GSH levels comes from in vivo Magnetic Resonance Spectroscopic (MRS) studies and cerebrospinal fluid measurements or post-mortem quantification of glutathione. Early in vivo studies reported 27–52% GSH reduction [60,61,62] in established schizophrenia. For many subsequent MRS GSH studies, the anterior cingulate cortex (ACC) has been the chosen region of interest given its relevance to schizophrenia as well as the technical advantage (uniform field homogeneity, higher signal-to-noise, narrow spectral peak width, low probability of susceptibility artefacts) offered by a midline MRS voxel placement [63]. A recent synthesis of cross-sectional MRS studies demonstrated a small but significant GSH reduction (effect size = 0.26) in the ACC region in schizophrenia [64]. Interestingly, of the 12 studies that were included in this meta-analysis, only two reported significant GSH differences between patients and HCs [60,65], contributing to a modest reduction in GSH among patients. This small but significant GSH reduction in schizophrenia has been reported by two other meta-analyses, one restricted to ultra-high field 7T MRS studies (effect size = 0.21) [66] and the other including all central measures of GSH MRS, CSF and post-mortem samples (effect size = 0.26) [67].

Since 2018, several MRS studies on ACC GSH in schizophrenia have been published [68,69,70,71,72,73,74] (Table 1). While the largest study to date supports Das and colleagues’ report of GSH reduction in early stages of schizophrenia [74], smaller studies have found no differences in patients when compared to healthy controls. Interestingly, samples of acutely symptomatic, untreated first-episode patients report higher levels of ACC GSH than healthy controls [72,73] which is not seen in other post-acute samples beyond 1–2 years of illness that are likely to include more treatment-resistant subjects [68,71,74].

These observations present a more nuanced picture of intracortical GSH aberrations than what can be expected from preclinical studies or from the large effect of size reduction of peripheral GSH measures in schizophrenia.

The heterogeneity observed in MRS studies of ACC GSH in schizophrenia highlights two distinct possibilities. First, there are likely to be at least two subgroups of patients, one with notable GSH deficit, and the other with near-normal or supra-normal levels of GSH compared to healthy subjects. Both these subgroups are most likely to be apparent among the untreated first episode patients with more florid positive symptoms but less cognitive deficits and more favourable treatment response profile than more established cases. One of the earliest reports linking increased intracortical GSH with favourable treatment outcomes came from the study of Wood and colleagues [75]. They reported a 22% increase in medial temporal GSH levels in first episode psychosis; in a sub-sample from this study [76], treatment related increase in GSH was associated with a gain in global functioning scores. ACC GSH levels were not examined in their sample. We observed that in untreated first episode patients, an elevated ACC GSH at the time of presentation occurred in those with more florid disorganization [73] but most of these subjects responded briskly to the regular antipsychotic treatments over the next 6 months [69]. Help seeking subjects with subthreshold psychotic symptoms display better social and occupational functioning when ACC GSH levels are higher [77]. In the meta-analysis of Das and colleagues we noted a small but significant increase in ACC GSH in bipolar disorder, a phenotype that is often associated with better functional outcomes than schizophrenia [64]. Samples with more established schizophrenia with residual symptoms and impaired functioning are likely to have an over-representation of the subgroup with GSH deficit. Based on the reported effect-sizes of the observed GSH deficit across chronic and partially treatment-resistant samples [64,67], it is likely that only a subgroup of patients have a notable GSH-deficit. Nonetheless, this subgroup of patients with GSH-deficit are likely to be treatment-resistant; a recent 7-Tesla MRS study in first-episode psychosis reports a large effect-size (Cohen’s d = 0.83) reduction in ACC GSH in 32 treatment-resistant patients compared to 106 non-treatment-resistant patients [78].

The second possibility is that in a patient with schizophrenia, GSH levels may vary with the stage of illness. Acute, untreated symptomatic state may relate to higher levels of GSH than stable clinical states of schizophrenia. Increase in GSH level may mark a compensatory response to acute oxidative stress that may not be sustained as the illness progresses to a more persistent stage. Longitudinal studies of intracortical GSH are limited; most patient cohorts with repeated MRS acquisitions to date did not measure GSH [79,80,81,82,83]. To our knowledge, only two studies report on longitudinal GSH measurements in psychosis to date. In a sample of 21 subjects with first episode schizophrenia scanned at baseline with <3 days of lifetime antipsychotic exposure and followed after 6 months of antipsychotic treatment, GSH levels were highly stable (Mean (SD) at baseline =1.71 (0.36), at 6-months = 1.75 (0.23)) [84]. The same stability in GSH levels was also seen in the 10 demographically healthy controls scanned at two time points (Mean(SD) at baseline =1.64 (0.25), at 6-months = 1.63 (0.32)) [84]. In a larger sample of 38 patients with first episode psychosis (onset within 2 years) and 48 healthy controls followed up over 4 years [85], GSH levels was found to have a near zero change in all five studied brain regions (ACC, thalamus, DLPFC, centrum semiovale, and orbitofrontal cortex) over time, strongly arguing for a ‘trait-like’ stability of GSH levels compared to other metabolites. These observations do not rule out within-individual differences over time, and the possibility of an early excess in untreated state; but provide a strong support for the presence of a distinct subgroup with GSH-deficit that is over-represented in more established phase of schizophrenia, in association with reduced treatment responsiveness and poor functional outcomes.

## 4. Factors Contributing to the Putative Intracortical GSH Deficit

In the next two sections we focus on factors contributing to the putative intracortical GSH-deficit and the pathophysiological consequences of this deficit.

Preclinical studies discussed earlier insinuate the possibility of a constitutional defect in GSH synthesis in schizophrenia. Defective production of GSH due to reduced expression of the GSH synthesizing enzyme Glycine Cysteine Ligase (GCL), has been demonstrated in patient-derived cell culture studies [57]. The high-risk variant gene encoding GCL’s catalytic subunit (GCLC) with eight or nine GAG repeats as opposed to seven was seen in 36% of patients, but only 3% of healthy controls in some samples [57]. Later studies failed to replicate an association between high-risk GCLC variant and lower GSH levels in schizophrenia [68,71,86]. Thus, the high risk variants do not consistently indicate low intracortical GSH [62]. In the same vein, low GSH levels in post-mortem brain tissue of patients with schizophrenia are observed despite normal levels of GCL and GSH peroxidase-like protein [86]. Furthermore, large-scale genome-wide association studies (GWASs) to date have not observed aberrations in the GSH synthesis pathway in schizophrenia (reviewed by Ermakov et al. [87]). Thus, a constitutional defect in GSH synthesis, if present, may be limited to a small number of patients. In most others, intracortical GSH levels are likely influenced by indirect factors (e.g., transcriptional regulators or epigenetic factors regulating gene expression [87,88]) affecting redox status.

Environmental factors, in particular lifestyle factors, also determine intracortical GSH. In fact a number of lifestyle factors in patients with mental illnesses may transiently affect GSH levels [89,90]. For example, in youth with mood disorders [91], apparent intracortical GSH-deficits are mostly explained by lifestyle factors (alcohol or smoking) [89,92,93]. In depression, this early-life reduction in GSH levels [94,95] appears to normalise later (15.7% increase, 1 year after first-episode depression [96]), likely explaining the higher than expected levels in later life [97]. At present, the relative influence of lifestyle factors on intracortical GSH in schizophrenia is unknown.

While several studies have examined the influence of lifestyle factors on peripheral antioxidant markers [98,99], intracortical GSH levels do not always reflect the peripheral antioxidant status in schizophrenia. For example, while the levels of scavenging antioxidant enzymes such as glutathione peroxidase levels correlate positively with ACC GSH in healthy subjects, such an effect is inconsistent, and absent in patients [55,100]. Peripheral antioxidant deficit in schizophrenia is of several magnitudes larger than the central GSH reduction reported so far (effect sizes 1.02 vs. 0.26 [67]). Further, localised brain tissue changes, such as an increase in free water concentration in grey matter (likely an effect of neuroinflammation), can affect the intracortical GSH in schizophrenia [101].

Taken together, intracortical GSH-deficit may represent a ‘failure mode’ of the redox system occurring in some patients with schizophrenia, with multifactorial pathways (genetic and lifestyle-related) converging to result in a relative GSH-deficit state over the course of this illness.

## 5. Consequences of the Putative Intracortical GSH Deficit

Irrespective of its origins, the presence of a pervasive inability to counter oxidative stress with increased intracortical GSH is likely to produce multiple downstream consequences in schizophrenia. Preclinical studies reviewed above indicate that a GSH-deficit is likely to facilitate excitotoxic damage [34,35] affecting dendritic spines [36] and axonal stability [37] and disrupting myelin formation [38,39,40,41]. Myelin deficits in schizophrenia (termed *dys*myelination [102,103]) are likely multifactorial [104,105]; oxidative stress is hypothesized to preferentially affect prefrontal myelin-generating precursor cells [106,107,108], affecting cortical microcircuits in the early phase of psychosis [49]. In first episode schizophrenia, GSH levels correlate with white matter integrity [39]; its relationship with myelin content and microstructure of the grey matter is still unknown. Most of the recorded grey matter changes in schizophrenia occurs in the immediate post-onset phase [24,109,110], coinciding with the critical period of intracortical myelination [111,112,113]. Intracortical myelin mostly insulates parvalbumin containing interneurons [114] that are highly susceptible to developmental factors influencing redox balance [115] as well as GSH deficit and associated glutamate-mediated excitotoxicity [30,116]. In a subsample of patients reported by Pan and colleagues [73], we obtained quantitative intracortical myelin measurement and noted several regions where patients had lower intracortical myelin in the presence of higher glutamate, only when GSH levels were lower than the median (bilateral dorsolateral prefrontal cortex, right superior temporal, and left precentral gyrus and right subgenual ACC). Thus, higher ACC glutamate related to lower prefrontal intracortical myelin, only when ACC GSH levels were also lower; this indicates a gatekeeping role for GSH in glutamate-related *dys*myelination (Figure 1). Relating GSH-deficit at the onset to subsequent intracortical myelin changes will provide compelling proof for the downstream effects of antioxidant aberrations on the illness trajectory.

In terms of clinical and functional consequences of low intracortical GSH, we observed a predictive relationship between low GSH and delayed response to antipsychotics. For every 10% move towards the lower end of GSH levels, patients had seven additional days of non-response to antipsychotics [69]. Lack of early response is a critical indicator of long-term poor outcomes in schizophrenia [117,118,119]. Significant deficits in ACC GSH observed in patients with early stage of treatment resistance is highly suggestive of a poor outcome trajectory [78]. Correlational studies also relate lower GSH to higher residual symptom burden [65], negative symptoms [120], and cognitive deficits [74] in established cases of schizophrenia.

In summary, patients with intracortical GSH deficit are likely to display structural and functional features indicative of poor outcome trajectories in schizophrenia. Nevertheless, due to the lack of temporal information required to separate causes vs. consequences, it is not clear if low ACC GSH plays a causal role in persistent poor outcome trajectory across the course of schizophrenia [67]. Longitudinal follow-up studies linking baseline GSH measurement to later long-term functional outcome are required in this regard.

## 6. Glutathione—Glutamate Relationship

In healthy physiological states, ACC GSH and glutamate levels are tightly correlated [65,68,69]. This may be related to the co-dependency of intracellular GSH and glutamate synthetic processes, with GSH acting as a reservoir for glutamate [121,122]. If GSH synthesis is reduced by blocking glutamate to GSH conversion, this increases cytosolic glutamate levels and synaptic excitatory potential. On the other hand, if GSH to glutamate conversion is blocked, this reduces glutamate concentration and synaptic excitation. Thus, intracortical GSH levels influence the prevailing glutamatergic tone, and vice versa. This raises the question of whether the subgroup of patients with higher-than-normal GSH levels in early stages of psychosis have a concurrent increase in glutamate levels.

Among the untreated first episode patients, we indeed noted higher glutamate levels in those with higher GSH levels, though this relationship was weaker than in healthy subjects [69]. Several observations indicate opposing effects of GSH and glutamate on the course of schizophrenia. Higher intracortical glutamate levels relate to reduced social function [69,123] and poorer treatment response [81] in the early stages and reduced cortical grey matter [83,124,125]. On the other hand, higher GSH levels relate to faster treatment response [69], better functioning [77], and preserved cortical grey matter [101] (See also [126,127] for association with peripheral GSH). We recently observed an opposing influence of glutamate and GSH on the intrinsic connectivity within the dACC node of the Salience Network in first episode psychosis [72]. Thus, GSH and glutamate may covary in their levels but have contrasting effects on the course of schizophrenia.

In schizophrenia, although the exact nature of glutamate dysfunction in schizophrenia is yet to be clarified, meta- and mega-analyses of MRS data demonstrate a glutamatergic deficit state, at least in the prefrontal cortex [66,128,129]. Emerging observations implicate antipsychotics in progressive glutamate reduction in schizophrenia [128] but evidence in this regard is still inconclusive (see [84,129]). Interestingly, lower GSH levels co-occur with reduced glutamate in the chronic stage [65,68], especially in ultra-treatment-resistant patients [71]. Thus, a GSH-deficit phenotype may be characterised by concurrent reduction in glutamate.

To understand the concurrent glutamate and GSH reduction in established schizophrenia, several explanatory models can be invoked. First, this concurrent reduction may reflect a persistent reduction in neural activity. In a healthy physiological state, sustained neural activity inevitably increases mitochondrial oxidative stress; as a result, increase in GSH may occur to reduce free radical burden. This activity-dependent GSH increase has been observed in short time scales (over a few minutes) using functional MRS in several [130,131,132] but not all [133,134,135] studies to date. While some increase in free radical production is inevitable in physiological states [136], there is no convincing evidence that this is sufficient to increase GSH levels, in the absence of a specific disruption in mitochondrial activity [137]. Instead, the task-dependent increase in GSH in physiological states may reflect an increase in conversion of the excess synaptic glutamate released during sustained neural activity; this diversion to GSH-synthesis can in turn reduce the availability of the precursor glutamate and reduce further excitation in a homeostatic manner. The presence of a negative correlation between BOLD signal and task-related GSH increase [130] in healthy subjects further supports this view. In this context, sustained reduction of intracortical GSH in a subgroup of patients may reflect a pervasively low neural activity of a given brain region. Thus, in chronic stages of schizophrenia when sustained effortful activity is diminished, low GSH levels can be expected accompanying a glutamatergic deficit state.

Another equally viable explanation for concurrent GSH and glutamate deficit in schizophrenia focusses on the putative primacy of the glutamatergic deficit [138]. A reduction in glutamate transport in the mitochondria may disrupt mitochondrial function and cause an increase of free radical production [139] and the consequent adaptive increase in GSH consumption. Finally, a third factor such as altered resting-state cerebral blood flow may lead to mitochondrial dysfunction, leading to both an increase in free radical species followed by lowered GSH and concomitant glutamatergic deficit. Recent evidence supports the notion of impaired mitochondrial function (especially, mitophagy, the elimination of defective mitochondria) in schizophrenia [140], though its relationship with GSH and glutamate in schizophrenia requires further investigation.

In summary, the fate of GSH and glutamate are highly intertwined throughout the longitudinal trajectory schizophrenia. Delineating the putative functional relevance of intracortical GSH deficit requires concomitantly measuring glutamate to clarify its relationship, as well as tracking the functional outcomes and cumulative treatment exposure [67] in schizophrenia.

## 7. Treatment-Engagement and Stratification Markers (GSH) for Antioxidant Trials

An exciting clinical utility of prospectively identifying patients with GSH-deficit is the therapeutic possibility of correcting it. A number of compounds with the potential to correct the effects of GSH deficit are in the pipeline (Table 2) [44,76,141,142,143,144,145,146,147,148,149,150,151,152,153,154,155,156,157,158,159,160,161,162,163,164,165,166,167,168,169,170,171]. Of these, N-acetylcysteine has been shown to improve cognition and negative symptoms (6 RCTs) [172], though the effect size is modest. Antioxidant therapies are more likely to benefit patients with a central antioxidant-deficit. Reliable characterisation of the GSH-deficit phenotype is critical in this regard. Furthermore, while the antioxidant pipeline is promising [44,76,141,142,143,144,145,146,147,148,149,150,151,152,153,154,155,156,157,158,159,160,161,162,163,164,165,166,167,168,169,170,171], reversing cognitive/negative symptoms requires longer trials that are substantially difficult to complete. We need reliable markers of biological efficacy that indicate engagement of the mechanistic target; this will help overcome several obstacles in clinical translation (e.g., targeted in vivo assay, dose finding, estimating trial duration, understanding placebo response). While MRS ACC GSH measurement provides a marker for treatment-engagement [173], given its variation with the illness phase, it is unlikely to become a standalone aid in patient selection for long-term trials. More accessible behavioural readouts (delayed response to antipsychotics or poor social or occupational functioning) may help in patient selection.

In terms of neuroimaging-based stratification markers for long-term clinical trials, two promising approaches need further study. One is the use of measures reflecting the likely downstream effects of the GSH-deficit. Given the critical importance of restoring glutamate homeostasis by manipulating GSH levels, indices of glutamatergic dysfunction may prove to be useful indirect markers of the need for antioxidant trials. Using dynamic causal modelling (DCM) of resting state functional MRI, Limongi and colleagues linked higher ACC glutamate in untreated schizophrenia to a model of cortical disinhibition in the ACC-insula network [18]. Glutamate-related disinhibition in this network predicted computational parameters of cognitive dysfunction as well as social withdrawal [18], while GSH levels had a robust relationship with the state of excitation–inhibition imbalance in this assay, with higher GSH predicting reduced disinhibition, an opposite effect from glutamate. An antioxidant that reliably reduces such markers of glutamate-related cortical disinhibition (likely by increasing intracortical GSH) is likely to be ‘hitting the target’ relevant to schizophrenia.

Another potential stratification marker is the estimation of early ‘response’ in MRS GSH to antioxidants as an indicator of the likelihood of success of long-term treatment. In essence, this is similar to the use of early symptom reduction as a longer-term prognostic indicator. Several antioxidants act by supplying the precursor for GSH synthesis, taking a longer time to increase intracortical GSH. For example, NAC increases intracortical GSH by 23% in schizophrenia only after 24 weeks but not immediately after administration (single dose effect = 1.3% increase [176]). In contrast, some antioxidants such as sulforaphane have a more rapid and stable effect on GSH levels [177,178] by activating the Nrf2 gene, the most dominant regulator of antioxidant transcription pathways. Sulforaphane is 80% bioavailable, reaches peak plasma levels 1 h after oral ingestion, with first-order kinetics [t_1/2_ ~2.5 h, 60% renal excretion at 8 h [179], full washout recorded in 3–5 days [180]] and has already been shown to increase intracortical GSH (using ultra-high field 7T MRS) in healthy volunteers (24% in 7 days) [181]. It is one of the antioxidants whose pharmacokinetics have been well studied [182], and whose safety profile has been established in clinical trials [183,184,185] with four ongoing trials in schizophrenia, making it a suitable drug to evaluate markers of biological efficacy. A notable sulforaphane-induced increase in MRS GSH level may predict a relative deficit state at the baseline, and thus a superior long-term functional response in patients.

## 8. Challenges and Opportunities

In summary, intracortical GSH is not abnormal in all patients who initially present with psychosis; but in a latent subgroup of patients with particularly adverse outcomes of schizophrenia, a pervasive intracortical GSH-deficit may result from a confluence of risk factors. A question of great translational importance is whether we can identify the subgroup of patients who will develop GSH-deficit at the onset of illness. The studies reviewed above suggest that this subgroup cannot be identified simply on the basis of genotyping or peripheral antioxidant measurements alone [67]. We suggest the following approaches for a reliable characterization of a pervasive intracortical GSH-deficit subtype:The putative consequences of GSH-deficit in schizophrenia likely involve aberrant functional connectivity within key brain networks (e.g., the Salience Network for dorsal ACC GSH deficit), myelination as well as grey matter microstructure. Longitudinal multimodal imaging, preferably starting from untreated states, and experimental ‘perturb-and-measure’ approaches with pharmacological agents such as sulforaphane or NAC will provide the required temporal information to characterize a causal role for intracortical GSH on these features. This is essential to establish the biological construct validity of the GSH-deficit phenotype in vivo.Attrition of the inception cohort is an important challenge in longitudinal studies of early-stage psychosis. Multi-site involvement is likely to be of critical importance to overcome this issue.Subgroup identification based on continuous biological measures is a statistical challenge; a single cut-off value for clinical decisions may not readily emerge. To mitigate this, in addition to the use of growth mixture and clustering models, normative estimates of MRS GSH values and classification approaches to inform cutoff optimization may be required.Several potential confounders/mediators of intracortical GSH (lifestyle variables, genetic variants, antipsychotic/antidepressant exposure, duration of illness, and substance use) require careful quantification to establish a relationship with outcomes of interest.Isolated measures of intracortical GSH do not provide the context in which the observed reduction occurs; concurrent static or dynamic measurement of glutamate will provide the relevant information to study putative mechanistic changes. As of now, 7T-MR spectroscopy (MRS), with its attendant improvisation in signal detection hardware, pulse sequences, and spectral modelling, is positioned as the only human in vivo technique that can confidently isolate glutamate from other molecules and concurrently estimate glutathione resonance. Among the MRS studies specifically optimized for GSH detection, 7T studies [65,186] report higher effect size GSH reduction in schizophrenia compared to 3T [55,187].

Current methods for measuring human brain intracortical GSH levels via MRS in conjunction with other critically relevant metabolites of the cortical microcircuit such as glutamate and GABA currently have quantification precision in the range of 20–40%. The use of MRS methods optimized for one of these key metabolites improves the quantification precision below 10% for one metabolite at the detriment of others (see [188] for a detailed review). Some recent developments to study redox status in vivo utilizing hyperpolarized ^13^C N-acetyl cysteine [189,190] or thiol-water proton exchange saturation transfer are promising to extend our insights into the glutathione system. Ultimately, developing MRS simultaneously optimized to provide quantification precision better than 10% for GSH, glutamate as well as GABA with greater spatial precision will open opportunities to explore mechanistic interventions targeting the schizophrenia redox-dysregulation subtype.

On the translational front, characterizing a redox-dysregulation subtype of schizophrenia using GSH-centered imaging holds significant promise for early intervention. Unlike data-driven subtyping approaches that are currently prevalent in the field, the redox-dysregulation or GSH-deficit subtype is based on the prediction of varying psychopharmacological outcomes i.e., superior response to antioxidants/poor response to antipsychotics. This may open the possibility of a stratified approach to pharmacological intervention in one subgroup and may reduce the iatrogenic burden of blanket trials across an entire diagnostic group over long time periods. Identifying reliable peripheral proxies for intracortical GSH will move this quest even closer to our clinics [191,192].

## 9. Conclusions

The GSH-deficit hypothesis offers a clinically actionable prognostic model in schizophrenia with a well-defined therapeutic utility. Longitudinal multimodal imaging studies combined with experimental ‘perturb-and-measure’ approaches can help delineate a putative redox-dysregulated subtype and establish its mechanistic primacy in the long-term trajectory of schizophrenia. This approach, if successful, will be a decisive step towards non-dopaminergic early intervention in schizophrenia.

## Figures and Tables

**Figure 1 antioxidants-10-01703-f001:**
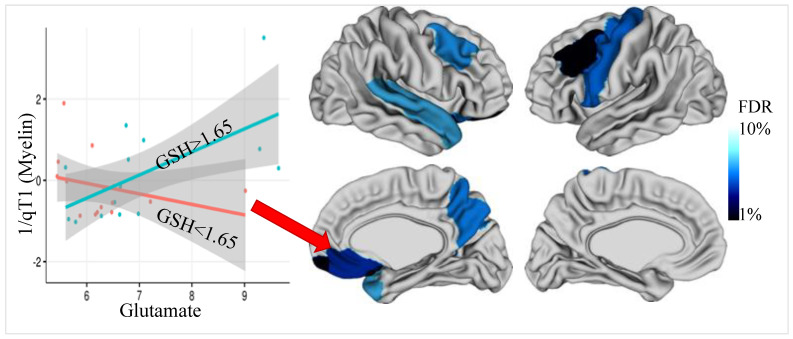
Permissive role of GSH on glutamate-related demyelination. Intracortical myelin (qT1) was mapped at the mid-cortical surface (50% depth between the white matter on pial surfaces) using the CIVET pipeline (https://github.com/aces/CIVET (accessed on 24 October 2021)) with T1-weighted images. Median qT1 values were gathered over cortical regions defined by the Desikan–Killiany Atlas (DKT40). In First Episode Psychosis (*n* = 30), glutamate relates to reduced prefrontal, orbitofrontal, and superior temporal intracortical myelin (1/qT1) only when GSH levels are also lower (red slope) in the dorsal ACC [MRS voxel not shown]—results were significant in 8 cortical regions after FDR correction (10%). Scatterplot shows the right lateral orbitofrontal cortex [contiguous with the medial surface that is shown on display], indicating significant glutamate-by-GSH interaction effect (t_24_ = 5.59, *p* = 9.52 × 10^−6^) on qT1, accounting for age and sex as covariates in multiple linear regression. Mean age 23.13 (SD 5.30), with 24/6 male/female subjects.

**Table 1 antioxidants-10-01703-t001:** Studies of intracortical GSH published since Das et al., 2018.

Study	No. Patients/Controls	Females/Males Patients	No. of Females/Males Controls	Age of Patients (Years) Mean (SD)	Age of Controls (Years) Mean (SD)	Clinical Features	Duration of Illness (Years) Mean (SD)
Coughlin et al., 2021	46/50 (16/10 *)	12/34	16/34	34.17 (11.8)	32.06 (11.28)	Chronic, stable phase of schizophrenia; 13% no APD. ACC GSH patients = HC	12.36 (11.45)
Dempster et al., 2020 ^a^	26/27	5/21	10/17	24.04 (5.4)	21.48 (3.57)	Acute, untreated psychosis; dACC GSH patients = HC; Higher GSH in patients with faster response to APD.	0.54 (1.25)
Godlewska et al., 2021	17/18 (14/18 *)	0/17	0/18	25.6 (1.1)	27.1 (0.8)	Stabilized first-episode; diagnostic information N/A; 12% no APD; ACC GSH patients = HC	2.54 (0.28)
Iwata et al., 2021 ^b^	21/26	5/16	7/19	46.3 (12.7)	40.8 (13.2)	First line treatment responders; dACC GSH patients = HC	20.0 (12.2)
Iwata et al., 2021 ^b^	27/26	8/19	7/19	40.5 (11.2)	40.8 (13.2)	TRS—Clozapine responders; dACC GSH patients = HC	16.4 (9.7)
Iwata et al., 2021 ^b^	24/26	5/19	7/19	44.8 (13.2)	40.8 (13.2)	TRS—Clozapine non-responders; dACC GSH patients = HC	23.5 (13.2)
Limongi et al., 2021 ^a^	19/20	7/12	9/11	21.7 (3.3)	21.3 (3.7)	Acute, untreated psychosis; 60% no APD; dACC GSH patients > HC	1.1 (1.8)
Pan et al., 2021 ^a,b^	16/25	3/13	11/14	21.81 (3.17)	22.12 (3.54)	Acute psychosis with high disorganization; >65% schizophrenia. dACC GSH patients > HC	0.98 (1.13)
Pan et al., 2021 ^a,b^	24/25	6/18	11/14	23.71 (5.43)	22.12 (3.54)	Acute psychosis with low disorganization; >80% schizophrenia. dACC GSH patients = HC	0.91 (1.7)
Wang et al., 2019	81/91 (74/88 *)	24/57	49/42	22.3 [4.4]	23.3 (3.9)	Stabilized first-episode; <65% of sample had schizophrenia. dACC GSH HC > patients	1.27 (0.8)

* Final sample with available MRS glutathione (GSH) data; all demographic information refers to the original larger sample. ^a^ overlapping samples; ^b^ same healthy control samples, distinct patient samples, reported in the same manuscript. APD: Antipsychotic Drugs, dACC: Dorsal Anterior Cingulate Cortex. HC: Healthy control subjects, NA: Data not available; SD: Standard Deviation; TRS: Treatment Resistant Schizophrenia.

**Table 2 antioxidants-10-01703-t002:** Potential therapeutic agents that can alter intracortical GSH.

Drugs that Activate Nrf2-Mediated GSH Regulation	Drugs that Increase or Stabilise GSH Levels via Other Mechanisms
Sulforaphane (NCT02880462; NCT02810964; NCT01716858; NCT04521868)	N-acetylcysteine [172,174] (NCT02505477, NCT03149107)
Curcumin [154,155] (NCT02104752, NCT02298985)	Direct liposomal GSH [157] (NCT01967667)
Resveratrol [158,159]	Ebselen [44,146] (NCT03013400)
Quercetin [160,161] (NCT04063124),	Ethyl eicosopentanoic acid [76,145]
Genistein [162,163] (NCT01982578)	Glucose-dependent insulionotropic polypeptide [143]
Andrographolide [164]	Alpha-lipoic acid [144] (NCT03788759)
CXA-10 [165]	L-arginine [142] (NCT04054973)
Bardoxolone [171]	S-adenosylmethionine [149,150]
Omaveloxolone [166] (NCT02255435),	Sarcosine [167,168]
Sulforadex (SFX-01) [175] (NCT02614742)	Serine [147,148] (NCT04140773, NCT03711500)
Dimethylfumarate [152,153]—now used for MS	Telmisartan [141] (NCT03868839)
Luteolin [169,170]	Trehalose [156] (NCT02800161)

NCT numbers indicate selected ongoing or recently completed clinical trials in neuropsychiatric disorders. Nrf2 = Nuclear factor-erythroid factor 2-related factor 2.

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
