# Peer review of "Is There a Glutathione Centered Redox Dysregulation Subtype of Schizophrenia?"

_antioxidants, 2021, doi:10.3390/antiox10111703_

Round 1

Reviewer 1 Report

The Authors focus the manuscript (a review) on the analysis of the literature on Schizophrenia and on the pressing need to identify pathways of poor outcome that remain ‘modifiable’ after the symptom onset. The Authors focuses on intracortical glutathione (GSH), they discuss how to establish or not the persistence of GSH-deficit subtype and how to clarify the role of the antioxidant system in disturbing brain structure and connectivity in schizophrenia at early stages.

The review is well written and the Authors provide a wide and complete analysis of the argument, with appropriate references.

Major considerations

  1. Lines 110-113. Is the consideration given in the sentence the result of data obtained by your group or not? If so specify and if not give one or more references.
  2. About Table 1 what does it mean the presence of number 1 (superscript) in the table legend? There is no number 1 (superscript) in table 1

3 Lines 169-170. The sentence is not necessary at the end of this paragraph 3. I suggest to move it at the beginning of the paragraph 4.

  1. Lines 240-241. The Authors mention Figure 1, but in the manuscript the only figure reported is Figure 2
  2. About Table 2 there is no indications i.e. Title and it lack of a list of abbreviations.

Minor considerations

The text needs to be revised as regards the italic words, the space before the references, the use of square or round brackets, the use of abbreviations that are not always reported in full in the text, some repetitions etc.

Author Response

The Authors focus the manuscript (a review) on the analysis of the literature on Schizophrenia and on the pressing need to identify pathways of poor outcome that remain ‘modifiable’ after the symptom onset. The Authors focuses on intracortical glutathione (GSH), they discuss how to establish or not the persistence of GSH-deficit subtype and how to clarify the role of the antioxidant system in disturbing brain structure and connectivity in schizophrenia at early stages. The review is well written and the Authors provide a wide and complete analysis of the argument, with appropriate references.

We thank the reviewer for highlighting the timeliness of this work.

Major considerations

1. Lines 110-113. Is the consideration given in the sentence the result of data obtained by your group or not? If so specify and if not give one or more references.

These statements referred to data from studies displayed in Table 1. Some of these are from our own group. We have now added the appropriate references for this claim.

Interestingly, samples of acutely symptomatic, untreated first-episode patients report higher levels of ACC GSH than healthy controls[72,73] that is not seen in other post-acute samples beyond 1-2 years of illness that are likely to include more treatment-resistant subjects[68,71,74].

2. About Table 1 what does it mean the presence of number 1 (superscript) in the table legend? There is no number 1 (superscript) in table 1

This superscript has been removed. The asterisk refers to the final samples reported in each of the analyses listed.

3. Lines 169-170. The sentence is not necessary at the end of this paragraph 3. I suggest to move it at the beginning of the paragraph 4.

We have now moved the statement “In the next 2 sections we focus on factors contributing to the putative intracortical GSH-deficit and the pathophysiological consequences of this deficit” to the next section.

4. Lines 240-241. The Authors mention Figure 1, but in the manuscript the only figure reported is Figure 2

This discrepancy has now been addressed. We have only one figure and the caption has been corrected.

5. About Table 2 there is no indications i.e. Title and it lack of a list of abbreviations.

We have now provided a title, and a note explaining the abbreviations.

Minor considerations

The text needs to be revised as regards the italic words, the space before the references, the use of square or round brackets, the use of abbreviations that are not always reported in full in the text, some repetitions etc.

We have now expanded all acronyms on first use.  The typological errors have been checked and corrected when noted.

Reviewer 2 Report

The manuscript by Palaniyappan et al. suggests that a GSH deficit may be a core element in a subgroup of patients with schizophrenia, a concept which, in turn, highlights the relevance of oxidative stress in the establishment of the disorder.

The manuscript focuses on the methodologies required for evaluating the GSH concentration in different brain areas of living patients and supports the use of antioxidants as an early intervention.

The manuscript is well structured and well documented and may be useful for clinicians and for researchers investigating the origin of schizophrenia.

I have only two suggestions that may improve the paper.

-On lines 288-290 it is stated: In a healthy physiological state, sustained neural activity inevitably increases mitochondrial oxidative stress; as a result, increase in GSH may occur to reduce free radical burden. To my knowledge, such statement is not supported by literature. It is indeed true that a certain amount of free radicals are physiologically produced as a consequence of the mitochondrial activity, however, increased free radical production occurs in the presence of disrupted or hampered mitochondrial activity, as it is well explained by M.P. Murphy (PMID: 19061483). Thus, one is allowed to look at the relationship “lower GSH levels - reduced glutamate” also in the opposite way with respect to the Authors’ view, i.e. reduced glutamate might lower GSH levels rather than the opposite, since reduced glutamate transport in the mitochondria may cause an increase of ROS production and the consequent consumption of GSH. In alternative, altered resting-state cerebral blood flow may lead to mitochondrial dysfunction, consequent increase of ROS and lowered GSH, as well as to concomitant decrease of glutamate.

-Authors may have missed a recent article (PMID: 34439564), discussing how oxidative stress may alter GABAergic signaling in many brain disorders, including schizophrenia. Please evaluate whether the content of that review may add valuable information to the present paper.

Author Response

The manuscript by Palaniyappan et al. suggests that a GSH deficit may be a core element in a subgroup of patients with schizophrenia, a concept which, in turn, highlights the relevance of oxidative stress in the establishment of the disorder. The manuscript focuses on the methodologies required for evaluating the GSH concentration in different brain areas of living patients and supports the use of antioxidants as an early intervention. The manuscript is well structured and well documented and may be useful for clinicians and for researchers investigating the origin of schizophrenia.

We thank the reviewer for highlighting the translational value of this review

I have only two suggestions that may improve the paper.

-On lines 288-290 it is stated: In a healthy physiological state, sustained neural activity inevitably increases mitochondrial oxidative stress; as a result, increase in GSH may occur to reduce free radical burden. To my knowledge, such statement is not supported by literature. It is indeed true that a certain amount of free radicals are physiologically produced as a consequence of the mitochondrial activity, however, increased free radical production occurs in the presence of disrupted or hampered mitochondrial activity, as it is well explained by M.P. Murphy (PMID: 19061483). Thus, one is allowed to look at the relationship “lower GSH levels - reduced glutamate” also in the opposite way with respect to the Authors’ view, i.e. reduced glutamate might lower GSH levels rather than the opposite, since reduced glutamate transport in the mitochondria may cause an increase of ROS production and the consequent consumption of GSH. In alternative, altered resting-state cerebral blood flow may lead to mitochondrial dysfunction, consequent increase of ROS and lowered GSH, as well as to concomitant decrease of glutamate.

The reviewer makes a very important point on the physiology of Glu-ROS-GSH relationship. We have incorporated these suggestions in the revised version. The added text is shown below.

While some increase in free radical production is inevitable in physiological states[1], there is no convincing evidence that this is sufficient to increase GSH levels, in the absence of a specific disruption in mitochondrial activity[2]

Another equally viable explanation for concurrent GSH and glutamate deficit in schizophrenia focusses on the putative primacy of the glutamatergic deficit[3]. A reduction in glutamate transport in the mitochondria may disrupt mitochondrial function and cause an increase of free radical production [4] and the consequent adaptive increase in GSH consumption. Finally, a third factor such as altered resting-state cerebral blood flow may lead to mitochondrial dysfunction, leading to both an increase in free radical species followed by lowered GSH and concomitant glutamatergic deficit. Recent evidence supports the notion of impaired mitochondrial function (especially, mitophagy, the elimination of defective mitochondria) in schizophrenia [5], though its relationship with GSH and glutamate in schizophrenia requires further investigation.

-Authors may have missed a recent article (PMID: 34439564), discussing how oxidative stress may alter GABAergic signaling in many brain disorders, including schizophrenia. Please evaluate whether the content of that review may add valuable information to the present paper.

This is a highly relevant review. We have now cited this as follows:

Intracortical myelin mostly insulates parvalbumin containing interneurons[6] that are highly susceptible to developmental factors influencing redox balance[7] as well as GSH deficit and associated glutamate-mediated excitotoxicity[8,9].

  1. Panov, A.; Schonfeld, P.; Dikalov, S.; Hemendinger, R.; Bonkovsky, H.L.; Brooks, B.R. The Neuromediator Glutamate, through Specific Substrate Interactions, Enhances Mitochondrial ATP Production and Reactive Oxygen Species Generation in Nonsynaptic Brain Mitochondria *. J. Biol. Chem. 2009, 284, 14448–14456, doi:10.1074/jbc.M900985200.
  2. Murphy, M.P. How Mitochondria Produce Reactive Oxygen Species. Biochem. J. 2009, 417, 1–13, doi:10.1042/BJ20081386.
  3. Adams, R.A.; Pinotsis, D.; Tsirlis, K.; Unruh, L.; Mahajan, A.; Horas, A.M.; Convertino, L.; Summerfelt, A.; Sampath, H.; Du, X.M.; et al. Computational Modeling of Electroencephalography and Functional Magnetic Resonance Imaging Paradigms Indicates a Consistent Loss of Pyramidal Cell Synaptic Gain in Schizophrenia. Biol. Psychiatry 2021, S0006-3223(21)01499–2, doi:10.1016/j.biopsych.2021.07.024.
  4. Hillen, A.E.J.; Heine, V.M. Glutamate Carrier Involvement in Mitochondrial Dysfunctioning in the Brain White Matter. Front. Mol. Biosci. 2020, 7, 151, doi:10.3389/fmolb.2020.00151.
  5. Khadimallah, I.; Jenni, R.; Cabungcal, J.-H.; Cleusix, M.; Fournier, M.; Beard, E.; Klauser, P.; Knebel, J.-F.; Murray, M.M.; Retsa, C.; et al. Mitochondrial, Exosomal MiR137-COX6A2 and Gamma Synchrony as Biomarkers of Parvalbumin Interneurons, Psychopathology, and Neurocognition in Schizophrenia. Mol. Psychiatry 2021, 1–13, doi:10.1038/s41380-021-01313-9.
  6. Micheva, K.D.; Wolman, D.; Mensh, B.D.; Pax, E.; Buchanan, J.; Smith, S.J.; Bock, D.D. A Large Fraction of Neocortical Myelin Ensheathes Axons of Local Inhibitory Neurons. eLife 2016, 5, e15784, doi:10.7554/eLife.15784.
  7. Abruzzo, P.M.; Panisi, C.; Marini, M. The Alteration of Chloride Homeostasis/GABAergic Signaling in Brain Disorders: Could Oxidative Stress Play a Role? Antioxid. Basel Switz. 2021, 10, 1316, doi:10.3390/antiox10081316.
  8. Lewis, D.A.; Curley, A.A.; Glausier, J.R.; Volk, D.W. Cortical Parvalbumin Interneurons and Cognitive Dysfunction in Schizophrenia. Trends Neurosci. 2012, 35, 57–67, doi:10.1016/j.tins.2011.10.004.
  9. Stedehouder, J.; Kushner, S.A. Myelination of Parvalbumin Interneurons: A Parsimonious Locus of Pathophysiological Convergence in Schizophrenia. Mol. Psychiatry 2016, doi:10.1038/mp.2016.147.